# Regio- and enantioselective nickel-alkyl catalyzed hydroalkylation of alkynes

Qian Gao[1,2], Wei-Cheng Xu[2], Xuan Nie [1], Kang-Jie Bian[2], Hong-Rui Yuan[2], Wen Zhang[2], Bing-Bing Wu [2] ✉ & Xi-Sheng Wang [1,2] ✉

The migratory insertion of metal-hydride into alkene has allowed regioselective access to organometallics, readily participating in subsequent functionalization as one conventional pathway of hydroalkylation, whereas analogous process with feedstock alkyne is drastically less explored. Among few examples, the regioselectivity of metal-hydride insertion is mostly governed by electronic bias of alkynes. To alter the regioselectivity and drastically expand the intermediate pools that we can access, one aspirational design is through alternative nickel-alkyl insertion, providing opposite regioselectivity induced by steric demand. Leveraging in situ formed nickel-alkyl species, we herein report the regio- and enantioselective hydroalkylation of alkynes with broad functional group tolerance, excellent regio- and enantioselectivity, enabling efficient route to diverse valuable chiral allylic amines motifs. Preliminary mechanistic studies indicate the aminoalkyl radical species can participate in metal-capture and lead to formation of nickel-alkyl, of which the migratory insertion is key to reverse regioselectivity observed in metal-hydride insertion.

In the past several years, nickel-catalyzed hydroalkylation of alkenes has been rapidly established as a versatile synthetic strategy to access functional molecules of synthetic interests, of which majority proceeded with Ni–H migratory insertion to afford the close-shell species, determining the regioselectivity and stereochemical control of this transformation[1–12]. However, the regio- and enantioselective hydro/functionalization of another prevalent feedstock hydrocarbons has remained rarely explored[13–19]. A breakthrough was made in 2018 by the Fu group, who reported the enantioconvergent hydroalkylation of symmetrical dialkyl alkynes and terminal alkyne with racemic secondary bromides under Ni catalysis in combination with triethoxysilane to yield the corresponding α-vinyl-substituted amides in good yield and excellent enantioselectivity[1]. Few reports on Co-H-catalyzed hydrosilylation, Cu-H-catalyzed hydroallylation and Ni-H-catalyzed hydrophosphination have been developed[20–22], among which the regioselectivities were all determined by metal-hydride insertion into triple bond, affording mostly benzylic functionalized products due to the electronic effect of aryl, alkyl-alkynes in previous reports.

Alternatively, if one metal-alkyl species is rendered instead, their migratory insertion to alkyne can be governed by steric demand between the alkyne substituents and incoming metal-alkyl species, possibly leading to opposite regioselectivity (Fig. 1a). Considering large number of accessible nickel oxidation state and postulated active species including Ni–H or Ni–alkyl that might be both involved in the reaction system, we reason that if a competitive Ni–alkyl insertion to alkyne could take place prior to Ni–H insertion, a distinctively regioselective hydroalkylation will be facilitated[11]. Notably, unlike migratory insertion to alkene where metal-hydride insertion is much easier than that of metal-alkyl, the alkyne insertion can bias both the thermodynamics and the kinetics, favoring metal-alkyl insertion. Moreover, the reverse β-hydride elimination would be kinetically much easier than β-alkyl elimination, suggesting another driving force of rapid migratory insertion of Ni-alkyl species formed in situ[23]. Due to close-shell migration insertion of Ni-alkyl to alkynes, the ligand can drastically affect the reaction outcome; with the use of chiral ligand, the initial C–C bond formation can be promoted in enantioconvergent

[1]Department of Pharmacy, The First Affiliated Hospital of USTC, Division of Life Sciences and Medicine, University of Science and Technology of China, Hefei, China. [2]Department of Chemistry, University of Science and Technology of China, Hefei, China. ✉e-mail: bbwuchem@mail.ustc.edu.cn; xswang77@ustc.edu.cn

**Fig. 1 | Motivation and design for insertion to nickel-alkyl bonds: regio- and enantioselective synthesis of allylic amines. a** Regioselectivity for asymmetric hydro/functionalization of internal alkynes: M−H insertion vs M−R insertion. **b** Our design for regio- and enantioselective hydroalkylation of internal alkynes: Ni-alkyl insertion. **c** Chiral allylic amines in biologically active molecules. **d** Known asymmetric catalysis strategies for synthesis of chiral allylic amines: limit scope and functional group compatibility. **e** This work: regio- and enantioselective hydroalkylation of internal alkynes via nickel-alkyl insertion.

manner, giving stereogenic allylic carbon. The alkenyl nickel intermediate can readily engage in sequential hydrogenation to access the enantioselective hydroalkylated products with an opposite regioselectivity (Fig. 1b). Indeed, while alkyl electrophiles are normally activated by nickel catalyst via single-electron-transfer path, the rapid combination of alkyl radical species with the nickel center is crucial to afford the high regioselectivity and enantioselectivity as we hypothesize, by obviating the unproductive radical C−C bond formation or Ni−H insertion that would corrode the regioselectivity.

Aliphatic amine serves an important structural motif that can be widely found in natural products and pharmaceuticals[24−26]. In particular, chiral allylic amines represent outstanding chiral building blocks en route to biologically active molecules (Fig. 1c)[27−31]. Over the past decades, tremendous efforts have been focused toward the development of efficient methodologies for their regio- and enantioselective synthesis from feedstock materials in a step-economical fashion, especially via asymmetric catalysis (Fig. 1d)[32−34]. Among the established methods of preparing chiral allylic amines, asymmetric allylic amination via π-allyl metal intermediate and asymmetric alkenylation of aldimines with organometallic reagents were the most visited transformations[35−46]. Nevertheless, regioselective allylic aminations normally required a sharp distinction of substituents on π-allyl metal intermediate with an aryl group or hydrogen atom installed on one side, and asymmetric alkenylations were dominated by the use of moisture- and air-sensitive organometallic reagent, impeding the practicality of the protocol. Indeed, asymmetric alkyne-imine reductive coupling via an oxidative cyclometallation has been developed as an efficient alternative solution[47−53], whereas arylsulfonyl-protecting group were required for both aryl imines by nickel catalysis and methylated alkynes by iridium catalysis for highly enantioselective

syntheses of allylic amines. To address these limitations and develop straightforward and efficient synthetic strategies, we sought inspiration from nickel-catalyzed radical hydroalkylation of alkynes with readily available amino acids as the aminoalkyl functionality precursor through decarboxylation, enabling the modular and versatile construction of various chiral allylic amines.

Putting the hypothesis into approach, herein, we develop a straightforward regio- and enantioselective hydroalkylation of alkynes for modular syntheses of chiral allylic amines. This hydroalkylation affords excellent regioselectivity and enantioselectivity, mild conditions, and high functional group tolerance. By identifying a nickel-alkyl insertion with internal triple bond, we are able to employ wide array of alkyne feedstocks and access distinctively different regioselectivity compared with conventional protocols initiated with nickel-hydride insertion (Fig. 1e).

## Results and discussion

**Optimization of reaction conditions.** With α-amino acid derivatives as the radical precursors, a nickel-catalyzed hydroalkylation of internal alkynes was explored. Our initial investigation commenced with 1-phenylpropyne (**2a**) as the pilot substrate and the NHPI ester of an α-amino acid (**1a**) as the radical precursor in the presence of a catalytic amount of NiBr₂·DME (10 mol%), Ca(OAc)₂ as the base, and (MeO)₃SiH as hydrogen source in DMAc at 40 °C (Table 1). While the reaction proceeded smoothly with pyridine-oxazoline **L1**, giving 32% yield, the desired chiral allylic amine (**3**) was obtained in low enantioselectivity (7% *ee*). Understanding the ligand might play a key role in enantiomeric control as well as accelerating reactivity, next, we carried out a careful screening of diverse oxazoline-based chiral ligands. Although tridentate Py-Box **L2** and bioxazoline **L3** exhibited very poor

**Table 1 | Nickel-catalyzed hydroalkylation of alkynes: optimization of conditions**

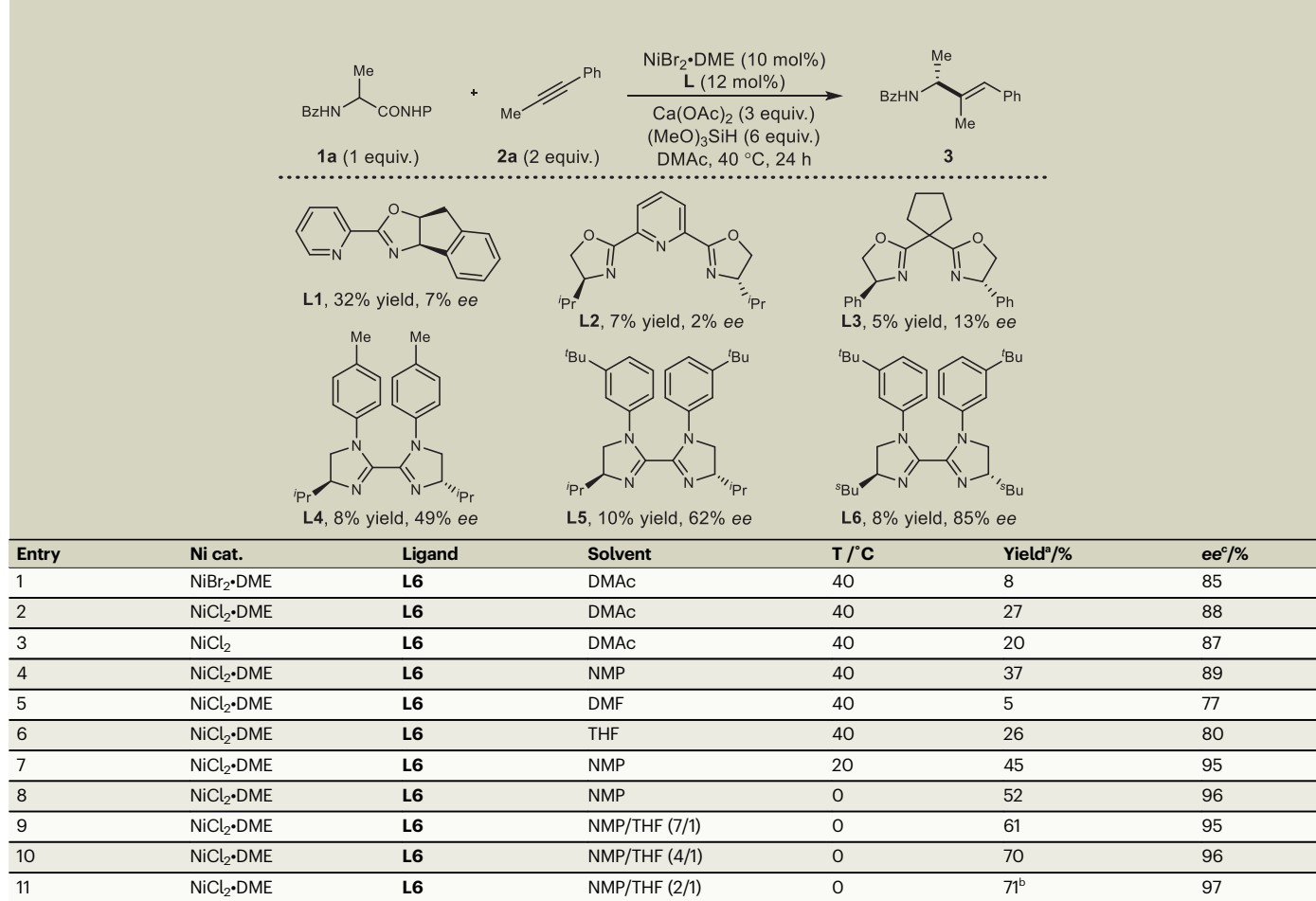

| Entry | Ni cat. | Ligand | Solvent | T /°C | Yield[a]/% | ee[c]/% |
|---|---|---|---|---|---|---|
| 1 | NiBr₂·DME | L6 | DMAc | 40 | 8 | 85 |
| 2 | NiCl₂·DME | L6 | DMAc | 40 | 27 | 88 |
| 3 | NiCl₂ | L6 | DMAc | 40 | 20 | 87 |
| 4 | NiCl₂·DME | L6 | NMP | 40 | 37 | 89 |
| 5 | NiCl₂·DME | L6 | DMF | 40 | 5 | 77 |
| 6 | NiCl₂·DME | L6 | THF | 40 | 26 | 80 |
| 7 | NiCl₂·DME | L6 | NMP | 20 | 45 | 95 |
| 8 | NiCl₂·DME | L6 | NMP | 0 | 52 | 96 |
| 9 | NiCl₂·DME | L6 | NMP/THF (7/1) | 0 | 61 | 95 |
| 10 | NiCl₂·DME | L6 | NMP/THF (4/1) | 0 | 70 | 96 |
| 11 | NiCl₂·DME | L6 | NMP/THF (2/1) | 0 | 71[b] | 97 |

*DMAc* dimethylacetamide, *NMP* N-methyl-2-pyrrolidone, *DMF* N,N-dimethylformamide, *THF* tetrahydrofuran, *ee* enantiomer excess.
[a]Isolated yields were given.
[b]Single r.r.
[c]The *ee* values were determined by HPLC on a chiral stationary phase.

enantioselectivities, to our delight, the employment of bisimidazoline **L4** as the ligand could remarkably improve the enantioselectivity of this transformation (49% *ee*), suggesting promising ligand framework that we can perform optimization of. In an effort to improve the enantioselectivity of the reaction, we subsequently explored a range of substituents on both the phenyl ring and the chiral carbons within the five-membered cyclic structure of the bisimidazoline ligands (for more details, see Supplementary Table 1). To our delight, the employment of bisimidazoline **L5**, which features a bulky substituent (*t*-Bu) placed at the *meta*-position on the phenyl ring to elevate steric hindrance, resulted in a notable enhancement of the enantioselectivity to 62% enantiomeric excess. Notably, the use of bisimidazoline **L6** as the chiral ligand in this hydroalkylation reaction, which installed on the chiral carbons of the five-membered cycle with a bulky group (*s*-Bu), furnished the enantioenriched allylic amine (**3**) in 85% *ee*, albeit with a lower yield of 8%.

With bisimidazoline **L6** as the optimal ligand, other parameters including nickel salts and solvents were then screened in detail. Different sorts of nickel sources were examined and NiCl₂·DME was the best choice to afford the allylic amine with 27% yield and 88% *ee* (Table 1, entries 1–3; for more details, see Supplementary Table 2). Subsequently, we screened a series of bases and silanes and found that Ca(OAc)₂ and (MeO)₃SiH still performed the best results (for more details, see Supplementary Table 3 and Table 4). With the investigation of the solvent effect, NMP stood out as the suitable solvent with a

slightly increased yield of 37%, however, DMF and THF afforded only relatively lower yields and enantioselectivities (Table 1, entries 4–6; for more details, see Supplementary Table 5). Next, we sought to lower the temperature to suppress side reaction of the hydride reduction, and delightedly, the yields and enantiomeric excess could be improved significantly (Table 1, entries 7–8; for more details, see Supplementary Table 6). Of note is that co-solvent could help with reaction efficiency significantly (Table 1, entries 9–11; for more details, see Supplementary Table 7), where we found in the solvent mixture of 2:1 NMP and THF at 0 °C, we were able to access this desired chiral allylic amine product in 71% of isolated yield, single r.r. and with up to 96% *ee* value. Notably, the exclusive *β*-selectivity products different from known reports[14–19] were obtained, suggesting the opposite regioselectivity that is governed by electronic bias between the alkyne substituents and incoming metal-alkyl species.

**Nickel-catalyzed regio- and stereoselective hydroalkylation of alkynes.** With the optimized reaction conditions in hand, the substrate scope of this nickel-catalyzed regio- and stereoselective hydroalkylation was next investigated (Fig. 2). Firstly, a series of arylalkyl alkynes were well compatible with this catalytic system for asymmetric hydroalkylation with NHP esters of an *α*-amino acid (**1**) in moderate to good yields and with high regio- and enantioselectivity (**3–21**). Besides phenylmethyl alkyne (**3**), fused ring derivatives such as naphthyl-methyl alkyne was also smoothly hydroalkylated to afford the desired

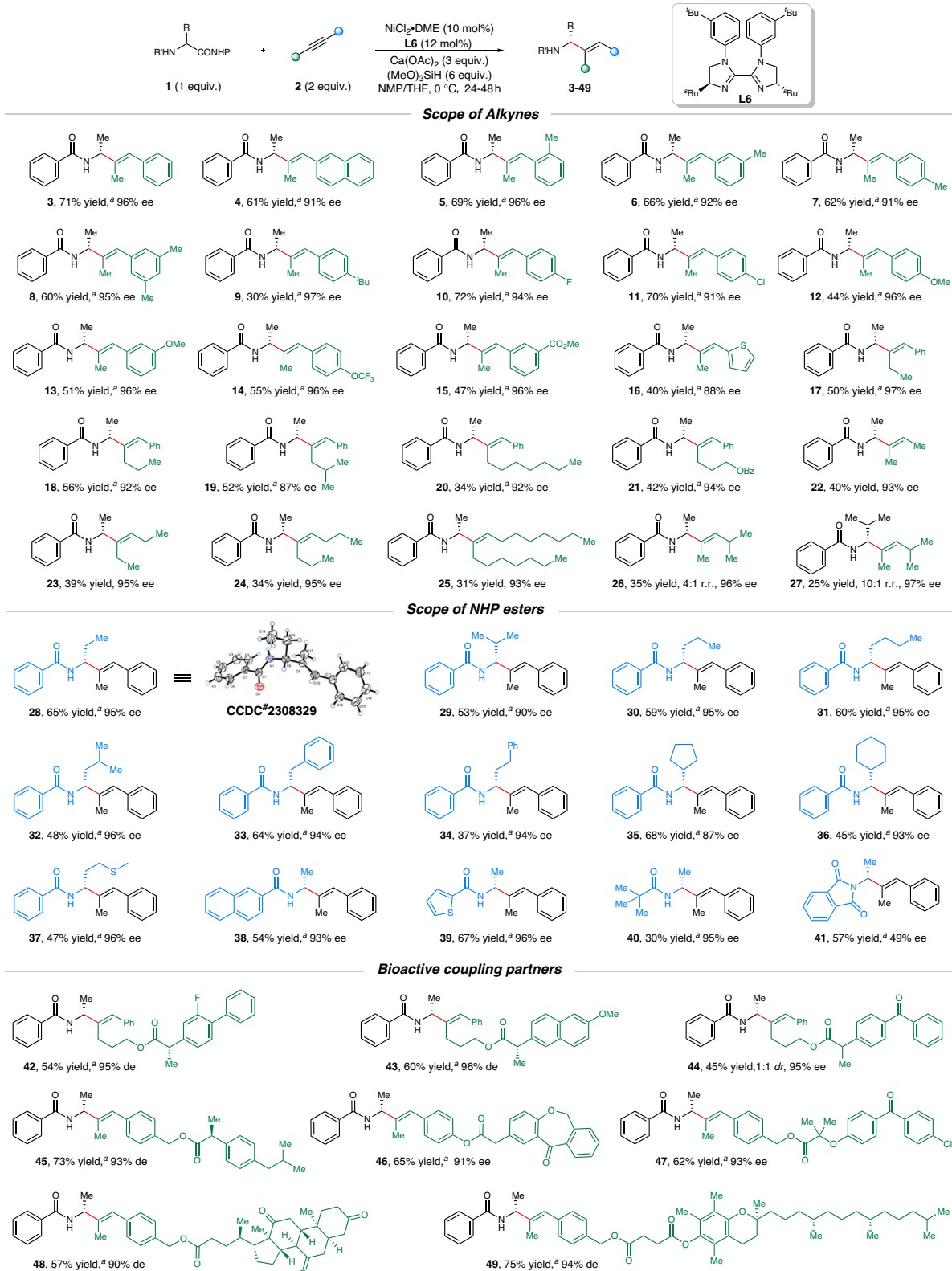

**Fig. 2 | Asymmetric synthesis of enantioenriched allylic amines.** [a]Single r.r.

chiral allylic amine (**4**) with single r.r. and 91% *ee* in 61% yield. Following the initial assessments, the impact of various substituents on the phenyl rings was explored. Specifically, both electron-donating substituents including methyl (**5**–**8**), *tert*-butyl (**9**), and methoxy (**12** and **13**), as well as electron-withdrawing substituents such as fluoro (**10**), chloro (**11**), trifluoromethoxy (**14**) and ester groups (**15**), were found to

be compatible within this catalytic framework. Worth noting is the fact that when alkynes featured methyl substituents at the *ortho*-, *meta*- or *para*-positions on their phenyl rings, the resulting products exhibited exceptional levels of both regioselectivity and enantioselectivity (**5**–**7**). Meanwhile, 3,5-disubstituented phenyl of the alkynes also behave well under this asymmetric catalytic system, giving the corresponding

chiral allylic amine (**8**) in moderate yield with excellent regio- and enantioselectivities (60% yield, single r.r. and 95% *ee*). Moreover, the arylmethyl alkynes installed with heteroarenes such as thiophene ring reacted efficiently in this reaction, affording the desired product (**16**) with a slightly lower yield and *ee*. Next, we moved on to explore the arrange of phenylalkyl alkynes of varying alkyl substitutions at the propargylic position (**17–20**). To our satisfaction, phenylalkyl alkynes containing simple alkyl chains without any heteroatoms could be well tolerated in this catalytic system, including the simplest ethyl (**17**), propyl (**18**), or seven-membered (**20**) long-chain alkyl groups. Of note is that excellent regio- and enantioselectivity was maintained when the protecting group such as OBz (**21**) was installed on the simple alkyl chain (42% yield, single r.r. and 94% *ee*). On the other hand, symmetrical dialkyl alkynes (**22–25**) were examined in this hydroalkylation reaction and obtained relatively lower yields with high enantioselectivities. Besides, the regioselectivity might arise (4:1-10:1 r.r.) from the steric hindrance influence between unsymmetrical dialkyl alkynes and the NHP ester of an *α*-amino acid with the bulky substituents (**26** and **27**).

With respect to the NHP ester scope, a broad range of diverse *α*-amino acid derivatives were readily compatible in alkyne hydroalkylation reaction with good efficiency (**28–37**, 37–68% yields, single r.r., 87–96% *ee*'s). Noteworthily, the NHP ester of *α*-amino acids with the bulky substituents, such as isopropyl (**29**), 5- and 6-membered rings (**35** and **36**) were also suitable coupling partners, providing desired products in moderate yields and excellent regio- and enantioselectivity. Furthermore, different *N*-protecting groups of *α*-amino acid derivatives, including naphthyl (**38**), thiophene (**39**), and *tert*-butyl (**40**) could be efficiently incorporated into this protocol, while switching the protecting groups from Bz to NHP led to considerable loss of *ee* (**41**, 57% yield, single r.r., 49% *ee*). Encouraged by the intriguing findings from our initial studies, we proceeded to undertake the challenge of asymmetrically hydroalkylating a selection of molecules with known biological activity, such as (*S*)-flurbiprofen (**42**), (*S*)-naproxen (**43**), ketoprofen (**44**), (*S*)-ibuprofen (**45**), isoxepac (**46**), fenofibric acid (**47**), dehydrocholic acid (**48**) and vitamin *E* (**49**). To our delight, the reaction showcased moderate to good yields along with exceptional regio- and enantioselectivities in all cases, which highlighting the substantial applicability and prospects of this method for efficiently constructing chiral allylic amine derivatives of pharmaceuticals or drug candidates.

**Synthetic applications.** To further demonstrate the synthetic utility of this strategy, we carried out the gram scale reaction under the standard conditions and the coupling product (**3**) was obtained without apparent loss of yield or enantioselectivity (66% yield, 96% *ee*) (Fig. 3a). Subsequently, we conducted various derivatization explorations on chiral allylic amine (Fig. 3b). Firstly, the chiral allylic amine (**3**) could be easily transformed into chiral amino ketone (**50**) in 75% yield with 96% *ee*. Following this, the obtained product (**50**) was effectively advanced to chiral amino alcohol (**51**), with a productive yield of 85% and an even greater enantioselectivity of 97% *ee*. Epoxidation of the chiral allylic amine was next proceeded, affording corresponding epoxides (**52**) in 71% yield with 98% *ee*. Furthermore, we also tested deprotection and protection of chiral allylic amine (**3**). Surprisingly, the *N*-Bz group can be readily removed to afford enantioenriched allylic amine (**53**), which could be further transformed to *N*-Ac group protecting product (**54**) in 80% yield with 96% *ee*. Overall, these preliminary synthetic applications indicated that the chiral center of allylic amine could be well maintained, and we thought that further increasing molecular complexity through alkene difunctionalization would be helpful to synthesize a variety of structurally diverse chiral amine derivatives.

**Mechanism investigation.** To gain more insights into the mechanism of this transformation, several preliminary mechanistic studies were performed. As proposed in the known reports[54–57], there are two possible pathways in this nickel-catalyzed hydroalkylation of alkynes: (a) Ni−H insertion and (b)Ni-alkyl insertion. It was found that when aryl, alkyl-alkynes were used as the coupling partner, *α*-selectivity of nickel species was more favored than *β*-selectivity due to the electronic effect of alkynes and conjugate effect of aryl group (Fig. 4a). Therefore, in the nickel hydride pathway, the *α*-selectivity was obtained followed by nickel species oxidative radical capture; However, the product was delivered with an opposite regioselectivity undergoing Ni-alkyl insertion pathway. As shown, the exclusive *β*-selectivity products were obtained when reacting with different alkynes with Ph, 2-nap, or thiophene group were examined, which was consistent with Ni-alkyl insertion. Moreover, the postulation could be further supported by regioselective coupling with a sterically biased internal alkyne, 4-methylpent-2-yne (Fig. 4b). Under the nickel hydride mechanism, the regioselectivity could be independent of the size of the alkyl radical moiety while the reverse regioselectivity was observed due to non-bonding steric effect between nickel-alkyl species and alkyne substituents[54]. Indeed, a positive correlation was observed between alkyl partner size and regioselectivity of hydroalkylation, which is completely consistent with a regioselectivity-determining Ni-alkyl insertion event.

Several deuterium labeling experiments were conducted to probe the origin of hydride source in the present hydroalkylation reaction. We firstly performed crossover experiment (Fig. 4c) with NHPI ester **1a** and alkyne **2a** as the substrates, deuterium-labeled PhSiD$_3$ as hydride source, and found that deuterium-containing product (**56**) was obtained in 48% yield and (**57**) was not detected at all. In contrast, deuterated product (**57**) was obtained albeit with a lower yield in the absence of alkyne **2a**, suggesting that reductive elimination from a possible H-Ni-alkyl intermediate might produce the hydrogenation product[58,59]. In addition, the subjection of D$_2$O into the standard conditions as a quenching reagent afforded only non-deuterated product (**3a**) but none of the deuterium-containing product (**56**), ruling out the possibility of protodemetalation[54]. Furthermore, the allylic amine (**3a**) was not detected without the addition of silane under the standard conditions (Fig. 4e, entry 1). All these results again suggest the origin of hydride source comes from the silicon hydride, and in situ-generated Ni-alkenyl species might undergo facile ligand exchange with hydride from silane, instead of protonated with water, showcasing mechanistic divergence with MacMillan's work[54].

The radical clock experiments were also conducted (Fig. 4d). In the presence of a stoichiometric amount of radical inhibitor TEMPO (2,2,6,6-tetramethyl-1-piperidinyloxy), the hydroalkylation reaction was completely inhibited and the TEMPO-radical adduct (**58**) was detected by HRMS. A competition experiment was conducted by adding 1.0 equiv. of electron-deficient alkene to the standard conditions and led to the formation of the adduct product (**59**). In addition, the experiment was performed with a cyclopropyl-containing NHPI ester (**60**) and alkyne, the ring-opened product (**62**) was observed, further indicating the intermediacy of *α*-aminoalkyl radicals. Moreover, the standard reaction proceeded smoothly in the dark (Fig. 4e, entry 2), demonstrating that the reaction is non-essential of photoinduced alkyl radical process. On the other hand, the metal Ni$^{I}$ species was suggested to trigger the reaction via the mixture of Ni$^{0}$ and Ni$^{II}$ catalyst, likely through disproportion, (Fig. 4e, entries 5 and 6), while it is not reactive when adding the Ni$^{0}$ catalyst alone (Fig. 4e, entries 3 and 4).

Based on the literatures[11,54–62] and our experiments, Ni$^{I}$X (**A**) is suggested to be the initial catalyst for the catalytic cycle (Fig. 4f). Under the reaction conditions, Ni$^{I}$X (**A**) reacts with an NHPI ester to form Ni$^{II}$XX' (**B**)and Ni$^{II}$(alkyl)X intermediate (**C**)[63,64]. Following the migratory insertion of Ni-alkyl intermediate to alkyne substrate, the Ni$^{II}$(alkenyl)X intermediate (**D**) will undergo ligand exchange with silane to form the Ni−H species Ni$^{II}$(alkenyl)H (**E**). Finally, reductive elimination leads to the formation of the coupling product (**F**) and the

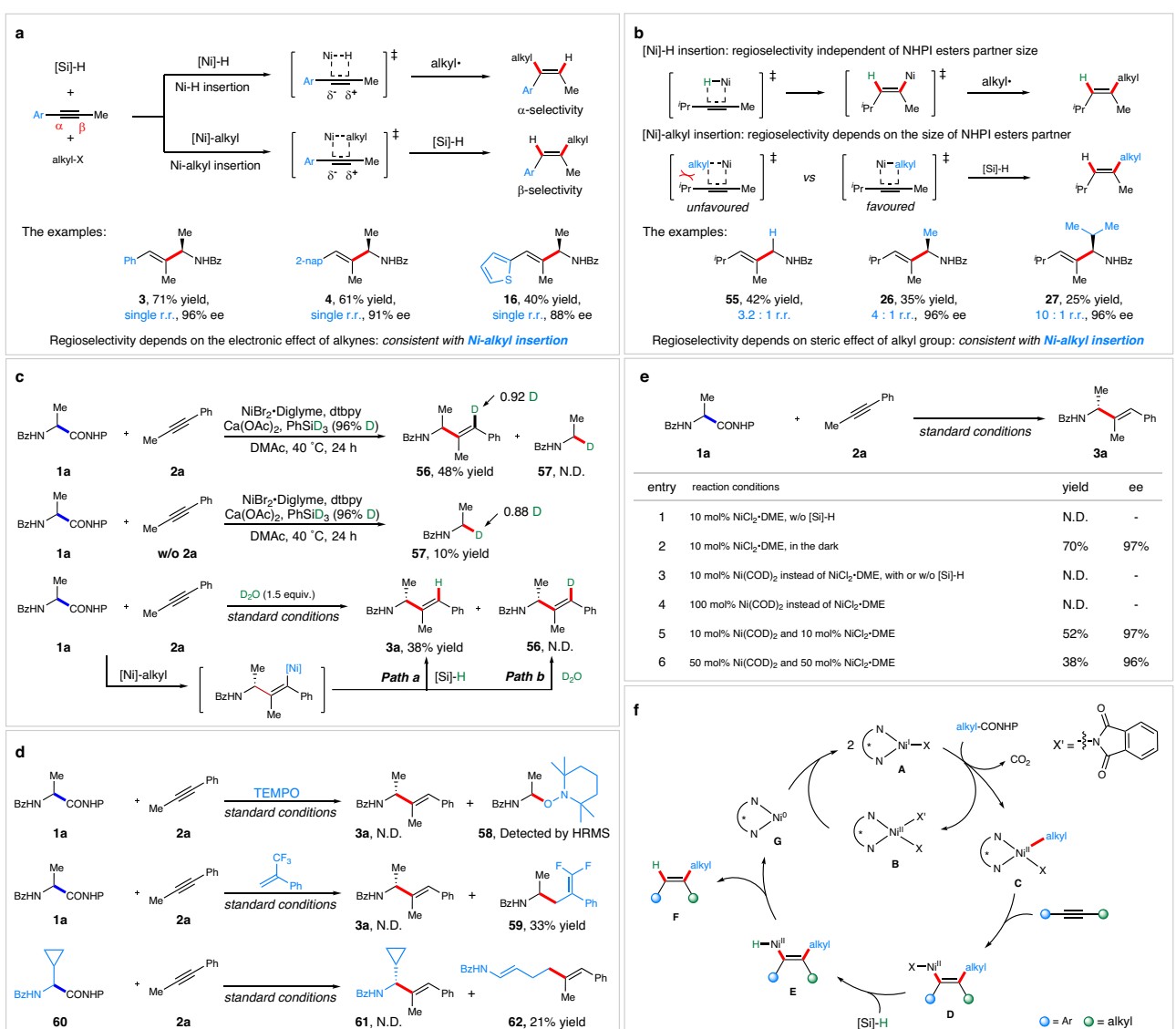

**Fig. 3 | Synthetic applications. a** Gram scale reaction. **b** Synthesis of diverse chiral amine derivatives.

**Fig. 4 | Mechanism investigation. a** Regioselectivity-determining step of aryl, alkyl-alkynes. **b** Regioselectivity-determining step of alkyl,alkyl-alkynes. **c** Deuterium labeling experiments. **d** Radical clock experiments. **e** The control experiments. **f** Proposed mechanism.

Ni⁰ species (**G**), which reacts with the Ni$^{II}$XX' species (**B**) to give Ni$^{I}$X (**A**), entering the next catalytic cycle.

In conclusion, we have demonstrated the regio- and enantioselective hydroalkylation of alkynes, representing an efficient and highly selective strategy for versatile and modular construction of allylic amines. This nickel-catalyzed asymmetric hydroalkylation protocol with various readily available amino acid derivatives as the alkyl radical source allows the expedient synthesis of chiral allylic amines with a high level of regioselectivity and enantioselectivity and broad substrate scope. We anticipate that this unusual regioselective and enantioselective hydroalkylation, initiated by close-shell nickel-alkyl rather than nickel-hydride migratory insertion, will provides a complementary strategy for accessing regioselective control divergent to conventional nickel hydride catalysis. This asymmetric multicomponent transformation provides the possibility for further exploration towards expedient synthesis of more challenging chiral allylic compounds, thus enabling changes to the conventional allylic carbon-carbon formation in organic synthesis.

## Methods

### General procedure for regio- and enantioselective hydroalkylation of alkynes via nickel-alkyl insertion for modular synthesis of chiral allylic amines

In glove box, NiCl$_2$•DME (0.01 mmol, 10 mol%), **L6** (0.012 mmol, 12 mol%), Ca(OAc)$_2$ (0.3 mmol, 3.0 equiv.) and alkyl NHP ester (0.10 mmol, 1.0 equiv.) were combined in a 5 mL oven-dried sealing tube. The vessel was evacuated and backfilled with Ar (repeated for 3 times). Alkyne (0.20 mmol, 2.0 equiv.), (MeO)$_3$SiH (0.60 mmol, 6.0 equiv.) and NMP/THF (v/v = 2/1, 0.5 mL) were then added via syringe. The tube was sealed with a Teflon-lined cap and stirred at 0 °C for 48 h. The reaction mixture was then diluted with EtOAc (~20 mL) and filtered through a pad of celite. The filtrate was added brine (20 mL) and extracted with EtOAc (2 × 15 mL), the combined organic layer was dried over Na$_2$SO$_4$, filtrated and concentrated under vacuum. The residue was then purified by flash column chromatography to give desired product as a solid or oil.

## Data availability

All data needed to support the conclusions of this manuscript are included in the main text or supplementary information. Data supporting the findings of this manuscript are also available from the authors upon request. X-ray crystallographic data for **28** (CCDC 2308329) has been deposited at the Cambridge Crystallographic Data Center. Copies of the data can be obtained free of charge via www.ccdc.cam.ac.uk/data_request/cif.

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

## Acknowledgements

Financial support for this work was provided by the National Key R&D Program of China (2023YFA1507500 (X.-S. W.)) and the National Science Foundation of China (52203196 (X. N.), 22271264 (X.-S. W.), 22301293 (B.-B. W.)).

## Author contributions

X.-S.W. and B.-B.W. conceived and designed the experiments. X.-S.W. directed the project. Q.G. performed chemical experiments and prepared the supplemental information. W.-C.X., H.-R.Y., and W.Z. prepared several ligands and substrates. X.N. performed chemical experiments in the revision process. X.-S.W., K.-J.B., and B.-B.W. co-wrote the

manuscript. All authors discussed the results and commented on the manuscript.

## Competing interests

The authors declare no competing interests.
