## [Peer Review File · Nature Communications]

REVIEWER COMMENTS

Reviewer #1 (Remarks to the Author):

The manuscript by Wang, Wu and co-workers have developed the regio- and enantioselective hydroalkylation of alkynes via nickel-alkyl insertion for modular synthesis of chiral allylic amines. Compared with previous methods to afford chiral allylic amines, this protocol exhibits high regio- and enantioselectivity, mild conditions and excellent functional group tolerance. More importantly, it is also robust for late-stage modification of versatile amino acids and drugs under mild conditions. As cited in the article, the field has witnessed a rapid development in nickel-catalyzed alkenes hydroalkylation over the past few years, establishing it as a widely applicable technique for producing molecules of substantial importance in synthesis. This advancement is largely attributed to the prevalent mechanism involving Ni-H migratory insertion, which yields closed-shell intermediates that are instrumental in dictating both the regioselectivity and stereochemical precision of the transformation. However, the regio- and enantioselective hydro/functionalization of alkynes has remained rarely explored. This work presented by the authors nicely complemented this gap and proposed a mechanism different from the traditional Ni-H migration insertion process. Therefore, this transformation offers a straightforward route for the synthesis of a series of chiral allylic amines via nickel-alkyl insertion for asymmetric hydroalkylation of alkynes. Publication in Nature Communications is recommended after revisions and some comments/suggestions are shown below:

- 1). The scope of NHP esters is limited due to the guiding effect of N-protecting groups with carbonyl groups, the authors need to make some attempts for other directing groups, for instance, ester groups.
- 2). On the other hand, the authors need to make some attempts on the symmetrical diaryl alkynes as a contrast with the symmetrical dialkyl alkynes.
- 3). The type of base and silane are critical for the conversion efficiency in this reaction. If possible, the authors need to provide the appropriate screening process in the SI.
- 4). There are some spelling errors and format errors need to be addressed, for example:

Manuscript P2, row 9 " asymmetric"; P5, row 7 " arrange"; P6, row 6 " enantioselectivities"; P6, row 10 " compatible".

Please double-check them all for the manuscript and SI.

Reviewer #2 (Remarks to the Author):

The research article by Wu and Wang describes a simple approach to construct chiral allylic amines via nickel-catalyzed asymmetric hydroalkylation of alkynes. Although there are many studies on chiral

allylamines as mentioned in the article, the authors further exploit an efficient and highly selective strategy for versatile and modular construction of chiral allylic amines. This nickel-catalyzed asymmetric hydroalkylation protocol with various readily available amino acid derivatives as the alkyl radical source allows the expedient synthesis of chiral allylic amines with a high level of regioselectivity and enantioselectivity and broad substrate scope. Moreover, the authors disclosed the competitive process of Ni-alkyl and Ni-hydride migration insertion in the mechanism and indicated a competitive Ni-alkyl insertion to alkyne would take place prior to Ni-H insertion, which is different from the conventional Ni-hydride migration insertion to the unsaturated bonds reported by the previous literatures. The results of the comparative experiments in the mechanism are convincing, due to the regioselectivity depends on the bonding steric effect between nickel-alkyl species and alkyne substituents, which is completely consistent with a regioselectivity-determining Ni-alkyl insertion event. This provides a deeper understanding of the reaction mechanism for the nickel-catalyzed asymmetric hydroalkylation of the unsaturated bonds. Thus, I recommend this work to be published on Nature Communications after minor revisions.

1. Some discussion for the limitation of the substrate scopes in this reaction system is needed. The symmetrical dialkyl alkynes (22-25, Fig. 2) were examined in this hydroalkylation reaction and showed relatively lower yields with high enantioselectivities, how about the symmetrical diaryl alkynes?
2. Please confirm that all ligands involved in SI have been reported before. If there are unknown ligands, please provide NMR data.
3. Some format issues need to be checked. In the Table 1 of the manuscript, the percentage sign for yield and enantioselectivity is not written.
4. Literatures needs to be added. Authors claim: "However, the regio- and enantioselective hydro/functionalization of another prevalent feedstock hydrocarbons has remained rarely explored¹³⁻¹⁵." Indeed, this is a challenging goal yet, however some works are in the literature and should be cited. Here are some selected examples: J. Am. Chem. Soc., 2015, 137, 4932; J. Am. Chem. Soc., 2019, 141, 12464; Nat Commun. 13, 4518 (2022); Authors claim: "Based on the literatures^{11, 50-55} and our experiments, ...". Although the citations are sufficient, there is a lack of some reports on NHP esters to introduce how it functions in the reaction. Here are some selected examples: ACS Catal., 2021, 11, 1640; Angew. Chem. Int. Ed., 2023, 62, e202305889; Angew. Chem. Int. Ed., 2023, 62, e202315203.

Reviewer #3 (Remarks to the Author):

The manuscript by Wu and Wang presents nickel catalyzed enantioselective hydroalkylation of alkynes for the synthesis of allylic amines. A wide variety of amino acid derivatives and alkynes was employed. In general, good to high enantioselectivities were observed. The SI is well prepared. Despite it is a

noteworthy contribution in the field, I have following major concerns before the paper can be suitable for publication in Nature Communications.

(1) The pioneering work of Fu (ref. 1) also showed few examples of enantioselective hydroalkylation of alkynes. The authors should put these initial results in the Fig. 1 and highlight clearly in the introduction to provide better state-of the art to the readers. Along this line, the following paper needs to be cited: J. Am. Chem. Soc. 2022, 144, 30, 13961–13972.

(2) The authors employed $\text{Ca}(\text{OAc})_2$ as base. Generally, in nickel-hydride chemistry this is not a very commonly employed base. What are the results with other bases like fluoride, phosphate or carbonate bases. Is $\text{Ca}(\text{OAc})_2$ exclusively the best base? If so, then authors should comment on this.

(3) What is the result with terminal alkene? This is an important aspect to explore.

(4) Along this line, did the authors try terminal or internal alkenes? Can the method be extended to this type of coupling partners?

(5) Late-stage functionalization of chiral compounds will be important to improve the quality of the paper. The authors should show derivatization of the obtained chiral compounds to synthetically useful building blocks.

Reviewer 1:

Question 1: The scope of NHP esters is limited due to the guiding effect of N-protecting groups with carbonyl groups, the authors need to make some attempts for other directing groups, for instance, ester groups.

Answer: We thank the helpful suggestion from the reviewer. We have used the NHP esters with Cbz- or Boc-protecting groups instead of Bz-protecting group as the starting materials in the standard conditions, but it was failed to provide the target products, which indicated the NHP esters with Bz-protecting group were essential in this reaction (for more details, see Figure S0 in the SI).

Question 2: On the other hand, the authors need to make some attempts on the symmetrical diaryl alkynes as a contrast with the symmetrical dialkyl alkynes.

Answer: We thank the reviewer for this question. We also tried to use the symmetrical diaryl alkynes as the starting materials for comparison with the symmetrical dialkyl alkynes as suggested, unfortunately, this reaction system was not compatible with symmetrical diaryl alkyne such as diphenylacetylene (for more details, see Figure S0 in the SI).

Question 3: The type of base and silane are critical for the conversion efficiency in this reaction. If possible, the authors need to provide the appropriate screening process in the SI.

Answer: Thanks for the good suggestion. We have added the relevant conditional screening in the SI as suggested (for more details, see Table S3 and Table S4 in the SI).

Table S3:

entry	base	yield / %	ee / %	entry	base	yield / %	ee / %
1	Ca(OAc) ₂	27	88	7	NaHCO ₃	8	91
2	NaOAc	n.d.	-	8	KHCO ₃	n.d.	-
3	CsOAc	n.d.	-	9	Na ₂ HPO ₄	14	85
4	Na ₂ CO ₃	trace	90	10	K ₃ PO ₄ ·H ₂ O	7	93
5	K ₂ CO ₃	n.d.	-	11	CsF	n.d.	-
6	Cs ₂ CO ₃	n.d.	-	12	CaF ₂	11	86

Table S4:

entry	Silane	yield / %	ee / %
1	(MeO) ₃ SiH	27	88
2	(EtO) ₃ SiH	n.d.	-
3	(MeO) ₂ MeSiH	n.d.	-
4	(EtO) ₂ MeSiH	n.d.	-
5	Et ₃ SiH	n.d.	-
6	PMHS	trace	84

Question 4: There are some spelling errors and format errors need to be addressed, for example: Manuscript P2, row 9 "asymmetric"; P5, row 7 "arrange"; P6, row 6 "enantioselectivities"; P6, row 10 "compatible". Please double-check them all for the manuscript and SI.

Answer: We thank the reviewer for the observation and apologize for the errors. We have made corrections to these spelling errors and double-checked the manuscript and SI.

Reviewer 2:

Question 1: Some discussion for the limitation of the substrate scopes in this reaction system is needed. The symmetrical dialkyl alkynes (22-25, Fig. 2) were examined in this hydroalkylation reaction and showed relatively lower yields with high enantioselectivities, how about the symmetrical diaryl alkynes?

Answer: We appreciate the suggestion! We have added some discussion for the limitation of the substrate scopes in this reaction system in the SI. The reaction system is not compatible with NHP esters with bulky substituents, such as *t*-Bu group and symmetrical diaryl alkyne, such as 1,2-diphenylethyne (for more details, see Figure S0 in the SI).

Question 2: Please confirm that all ligands involved in SI have been reported before. If there are unknown ligands, please provide NMR data.

Answer: We thank the reviewer for this question. We have double-checked the data for all the ligands and made sure all of them have been reported before.

Question 3: Some format issues need to be checked. In the Table 1 of the manuscript, the percentage sign for yield and enantioselectivity is not written.

Answer: We thank the reviewer for this observation and apologize for the typos. We have revised the mistake in the Table 1 of the manuscript.

Question 4: Literatures needs to be added. Authors claim: “However, the regio- and enantioselective hydro/functionalization of another prevalent feedstock hydrocarbons has remained rarely explored¹³⁻¹⁵.” Indeed, this is a challenging goal yet, however some works are in the literature and should be cited. Here are some selected examples: *J. Am. Chem. Soc.*, 2015, 137, 4932; *J. Am. Chem. Soc.*, 2019, 141, 12464; *Nat Commun.* 13, 4518 (2022); Authors claim: “Based on the literatures^{11, 50-55} and our experiments, ...” . Although the citations are sufficient, there is a lack of some reports on NHP esters to introduce how it functions in the reaction. Here are some selected examples: *ACS Catal.*, 2021, 11, 1640; *Angew. Chem. Int. Ed.*, 2023, 62, e202305889; *Angew. Chem. Int. Ed.*, 2023, 62, e202315203.

Answer: Thanks for the suggestion. We have cited the relevant literatures in the manuscript as suggested.

Reviewer 3:

Question 1: The pioneering work of Fu (ref. 1) also showed few examples of enantioselective hydroalkylation of alkynes. The authors should put these initial results in the Fig. 1 and highlight clearly in the introduction to provide better state-of the art to the readers. Along this line, the following paper needs to be cited: *J. Am. Chem. Soc.* 2022, 144, 30, 13961 – 13972.

Answer: We appreciate the constructive and helpful suggestion. In our manuscript, the aryl,alkyl-alkynes was using as the starting materials in this reaction shown in the Fig.1 and the excellent regioselectivities (>20:1) were obtained for the mostly benzylic functionalized products due to the electronic effect of aryl,alkyl-alkynes, which was relatively different from the Fu’s work. However, we strongly agreed with that Fu group have made the

pioneering work for the enantioselective hydroalkylation of alkynes. Therefore, we added and highlighted the statement in the introduction and cited the literature (J. Am. Chem. Soc. 2022, 144, 30, 13961-13972) shown in ref. 13 as suggested. The revised statement was also shown below:

A breakthrough was made in 2018 by the Fu group, who reported the enantioconvergent hydroalkylation of symmetrical dialkyl alkynes and terminal alkyne with racemic secondary bromides under Ni catalysis in combination with triethoxysilane to yield the corresponding α -vinyl-substituted amides in good yield and excellent enantioselectivity¹.

Question 2: The authors employed Ca(OAc)₂ as base. Generally, in nickel-hydride chemistry this is not a very commonly employed base. What are the results with other bases like fluoride, phosphate or carbonate bases. Is Ca(OAc)₂ exclusively the best base? If so, then authors should comment on this.

Answer: We thank the reviewer for the question. Actually, we have screened different types of bases including fluoride, phosphate and carbonate bases, however, the base of Ca(OAc)₂ exhibited the most excellent reaction performance. We have added some comments in the manuscript and shown the table for the screening bases in the SI as suggested (for more details, see Table S3 in the SI).

entry	base	yield / %	ee / %	entry	base	yield / %	ee / %
1	Ca(OAc) ₂	27	88	7	NaHCO ₃	8	91
2	NaOAc	n.d.	-	8	KHCO ₃	n.d.	-
3	CsOAc	n.d.	-	9	Na ₂ HPO ₄	14	85
4	Na ₂ CO ₃	trace	90	10	K ₃ PO ₄ ·H ₂ O	7	93
5	K ₂ CO ₃	n.d.	-	11	CsF	n.d.	-
6	Cs ₂ CO ₃	n.d.	-	12	CaF ₂	11	86

Question 3: What is the result with terminal alkene? This is an important aspect to explore.

Answer: We thank the reviewer for the question. We have tried some terminal alkenes as the starting materials, unfortunately, this reaction system was not compatible with terminal alkenes such as styrene or but-3-en-1-ylbenzene.

Question 4: Along this line, did the authors try terminal or internal alkenes? Can the method be extended to this type of coupling partners?

Answer: We thank the reviewer for the question. We have tried some terminal or internal alkenes, unfortunately, the experimental results indicated that alkenes are not suitable in this reaction system.

Question 5: Late-stage functionalization of chiral compounds will be important to improve the quality of the paper. The authors should show derivatization of the obtained chiral compounds to synthetically useful building blocks.

Answer: We appreciate the constructive and helpful suggestion. We carried out the gram scale reaction under the standard conditions (**Figure 3**) and the coupling product (**3**) was obtained without apparent loss of yield or enantioselectivity (66% yield, 96% ee). Moreover, we also conducted various derivatization explorations on chiral allylic amine and these preliminary synthetic applications indicated that the chiral center of allylic amine could be well maintained, and we thought that further increasing molecular complexity through alkene difunctionalization would be helpful to synthesize a variety of structurally diverse chiral amine derivatives.

REVIEWERS' COMMENTS

Reviewer #1 (Remarks to the Author):

The authors have addressed all my concerns and the manuscript is recommended to be accepted as it stands.

Reviewer #2 (Remarks to the Author):

In this revised manuscript, the authors have fully addressed my concerns in the first revision, and my view is that this manuscript is now suitable for publication.

Reviewer #3 (Remarks to the Author):

This revised manuscript addresses most of the concerns that I raised on the initial manuscript. In particular, the authors have provided a better state-of the art with up to date literature background. They have also shown derivatization of the obtained chiral compounds to synthetically useful building blocks. I recommend acceptance in Nature Communications.